# The Role of Extracellular Vesicles in Optic Nerve Injury: Neuroprotection and Mitochondrial Homeostasis

**DOI:** 10.3390/cells11233720

**Published:** 2022-11-22

**Authors:** Mira Park, Hyun Ah Shin, Van-An Duong, Hookeun Lee, Helen Lew

**Affiliations:** 1Department of Ophthalmology, CHA Bundang Medical Center, CHA University, Seongnam 13496, Gyeonggi-do, Republic of Korea; 2Gachon Institute of Pharmaceutical Sciences, Gachon College of Pharmacy, Gachon University, Incheon 21936, Republic of Korea

**Keywords:** human placenta derived mesenchymal stem cells (PSCs), optic nerve compression model, neuroprotection, mitochondria, mitophagy

## Abstract

Stem cell therapies hold great promise as alternative treatments for incurable optic nerve disorders. Although mesenchymal stem cells exhibit various tissue regeneration and recovery capabilities that may serve as valuable therapies, the clinical applications remain limited. Thus, we investigated the utility of extracellular vesicles (EVs) from human placenta-derived mesenchymal stem cells (hPSCs) in this context. Hypoxically preconditioned hPSCs (HPPSCs) were prepared via short-term incubation under 2.2% O_2_ and 5.5% CO_2_. The EVs were then isolated. R28 cells (retinal precursor cells) were exposed to CoCl_2_ and treated with EVs for 24 h. Cell proliferation and regeneration were measured using a BrdU assay and immunoblotting; ATP quantification revealed the extent of the mitochondrial function. The proteome was determined via liquid chromatography-tandem mass spectroscopy. Differentially expressed proteins (DEPs) were detected and their interactions identified. HPPSC_EVs functions were explored using animal models of optic nerve compression. HPPSC_EVs restored cell proliferation and mitochondrial quality control in R28 cells damaged by CoCl_2_. We identified DEPs (*p* < 0.05) that aided recovery. The mitochondrial DEPs included LONP1; PARK7; VDAC1, 2, and 3; HSPD1; and HSPA9. EVs regulated the levels of mitophagic proteins in R28 cells injured by hypoxia; the protein levels did not increase in LONP1 knockdown cells. LONP1 is a key mediator of the mitophagy that restores mitochondrial function after hypoxia-induced optic nerve injury.

## 1. Introduction

No current treatment handles irreversible optic nerve damage well; many scholars have sought to improve recovery of retinal ganglion cells (RGCs) [1]. All limbal stem cells, retinal pigment epithelial cells, mesenchymal stem cells (MSCs), and embryonic stem cells have been tested [2]. MSCs recruit immune cells by releasing immunomodulators and chemokines [3]. Additionally, secreted paracrines from MSCs have also been shown to be neuroprotective [3,4]. In models of ischemia, MSC injections increased the numbers of RGCs and the levels of brain-derived neurotrophic factor (BDNF), ciliary neurotrophic factor (CTNF), and the fibroblast growth factor (bFGF) [5]. Additionally, MSCs repair ischemia [6,7,8], and MSCs derived from bone marrow, human umbilical cord blood, dental pulp, and placenta (PMSCs) induce optic nerve regeneration and axonal growth [1,9]. However, the clinical applications of MSCs remain limited. Extracellular vesicles (EVs) of various cells have thus been tested. MSC EVs are small and prolific, and do not seem to induce tumor formation [10]. We thus focused on nano EVs. In a previous study, we showed that EVs of hPSCs afforded neuroregeneration after optic nerve compression (ONC); hPSC-derived EVs rescued retinal precursor cells affected by hypoxic injury via the WNT/β-catenin signaling pathway activated by UBA2 [11]. Hypoxia facilitates MSC maintenance by stabilizing various factors [12,13] that enhance stem cell function/survival, thus upregulating cellular differentiation and migration, increasing growth factor secretion, regulating intracellular transduction, and enhancing resistance to damage [14,15,16]. Several studies have reported that MSC immunomodulatory and regenerative properties improve by 1–5% O_2_ under hypoxic conditions [17,18,19]. Hypoxia enhances the effects of PMSCs on optic nerve regeneration and recovery [1]. In this study, we explored the therapeutic utility of human hypoxically preconditioned MSCs (HPPSCs) using hypoxia-damaged R28 cells and a rodent ONC model. Neurons are energy-demanding, and mitochondrial homeostasis is essential in terms of neuronal network formation, cell communication and activity; notably, unique molecular mechanisms regulate the mitochondrial distribution [20]. The importance of mitochondrial function/dysfunction in the context of neuronal injury is widely acknowledged. RGCs are very vulnerable to mitochondrial dysfunction because unmyelinated axons bend when exiting the eye and only then become myelinated (within the optic nerve); such cells exhibit the highest density of mitochondria in RGCs [21]. Sometimes, mitochondria are transported to regions of damage such as axotomized axons; a correlation is evident between such mobilization and axon regeneration [22,23,24]. Axonal mitochondria are not cleared via localized mitophagy and exhibit dramatic retrograde transport during mild stress [25]. Here, we investigated the neuroprotective roles played by functionally enhanced EVs from HPCs in in vitro and in vivo optic nerve injury models in an effort to develop a new therapy for patients with optic neuropathy.

## 2. Materials and Methods

### 2.1. Human Hypoxia-Preconditioned Mesenchymal Stem Cells (hHPPSCs) Preparation and Isolation of Extracellular Vesicles from hPSCs and hHPPSCs

Human placenta stem cells (hPSCs) were obtained from CHA Biotech (Seongnam, Republic of Korea). Human PSCs were maintained with minimum essential medium (MEM)-alpha GlutaMAX (Thermo Fisher Scientific, Waltham, MA, USA) supplemented with 10% FBS (Thermo Fisher Scientific), 1% penicillin/streptomycin (Thermo Fisher Scientific), 25 ng/mL human fibroblast growth factor 4 (Peprotech Inc., Rocky Hill, NJ, USA), and 1 μg/mL heparin (Sigma-Aldrich, St. Louis, MO, USA). To improve the recovery effect of hPSCs, hPSCs were incubated in 2.2% O_2_ and 5.5% CO_2_ conditions for 30 min. After exposure, the hPSCs and hHPPSCs were incubated in MEM-alpha GlutaMAX containing 10% exosome-free FBS (ThermoFisher Scientific) with supplements for 48 h. We isolated extracellular vesicles (EVs) from hHPPSCs (passages of 8–10) using ExoQuick-TC kit (SYSTEM BIOSCIENCES (SBI), Palo Alto, CA, USA). The experimental process followed the manual provided by the company. The exosome obtained in the final step was dissolved in 100 μL PBS and quantified by the BCA method. The EVs were stored at −80 ℃. Using PMX-120 ZetaView^®^ Mono Laser (Particle Metrix, Meerbusch, North Rhine-Westphalia, Germany), the size of EVs was measured.

### 2.2. Cell Culture and EVs Treatment

Immortalized R28 retinal precursor cells were maintained in Dulbecco’s minimal Eagle’s medium (DMEM; Sigma-Aldrich) with 10% fetal bovine serum (FBS; Thermo Fisher Scientific), 1× minimal essential medium (MEM) with nonessential amino acids (Thermo Fisher Scientific), 100 μg/mL gentamicin (Sigma-Aldrich), and 1% penicillin-streptomycin (Thermo Fisher Scientific). A hypoxic environment was created by using cobalt chloride (CoCl_2_) in cells. R28 cells were seeding in a number of 2 × 10^5^ and then treated with 200 μM of CoCl_2_ for 9 h. Then, EVs of a size of 170–180 nm were treated on the damaged R28 cell at a concentration of 12 μg/mL. After 24 h, the cells were harvested and prepared for analysis.

### 2.3. BrdU (Bromodeoxyuridine) for Cell Proliferation Analysis

Proliferation levels of R28 cells were determined using BrdU ELISA kit (CellBioLabs, Inc., San Diego, CA, USA). R28 cell culture was performed by replicating three times for each experiment on a 96-well plate. The experimental step was carried out according to the manufacturer’s protocol.

### 2.4. Determination of ATP Levels

To evaluate ATP production, cells lysates from damaged R28 cells were lysed. ATP levels were detected by the ATP determination kit (Thermo Fisher Scientific). The production levels were determined by lumenometer (Spectramax iD5; Molecular Devices, San Jose, CA, USA) according to the manufacturer’s instructions.

### 2.5. Reactive Oxygen Species (ROS) Measurements

Mitochondrial superoxide production, a type of reactive oxygen species (ROS) of damaged R28 cells, was measured by MitoSOX Red (Thermo Fisher Scientific). The cells seeded on the cover slip were treated with MitoSOX Red (5 µM) at 37 °C for 10 min protected from light, and then washed. The stained cover slips were mounted on slides using a mounting solution. Fluorescence was detected using a Zeiss LSM 880 confocal microscope (Carl Zeiss, Oberkochen, Baden-Württemberg, Germany). Using ZEN software (Carl Zeiss), the intensity of staining was measured.

### 2.6. Small-Interfering RNA for LONP1 Protein

For knockdown of Lonp1 protein, the target sequence of siRNA (Bioneer Corporation, Daejeon, Republic of Korea) used was as follows: siRNA rat Lonp1, 5′- GGG AAC GCU UAA AGG AGC UUG UGG U. Negative Control (Bioneer Corporation) was used as used as scramble. R28 cells were transfected using Lipofectamine 3000 (Thermo Fisher Scientific) according to the manufacturer’s instructions.

### 2.7. Immunoblot Analysis

The experiment step of immunoblotting has been described in previous study [1]. Briefly, total proteins from cell or tissue were extracted by a RIPA or a PRO-PREP buffer (iNtRON Biotechnology, Gyeonggido, Republic of Korea). Concentration of protein was determined by BCA method (Thermo Fisher Scientific). Target proteins were separated by SDS-PAGE and then transferred to PVDF membranes (GE Healthcare, Chicago, Il, USA). The membranes were incubated with anti-Hif-1α (ab179483), Gap43 (ab16053), Ermn (ab90893), Mul1 (ab209263; Abcam, Cambridge, UK), Vegf (GTX102643; GeneTex, Irvine, CA, USA), Thy-1 (SC-53116), Bnip3 (SC-56167), Pink1(SC-518052), Prdx5 (SC-133072), β-actin (SC-47778; Santa Cruz Biotechnology, Santa Cruz, CA, USA), Neurofilaments (#2837), Lc3b(#2775), Mfn2 (#9482), Prdx2 (#46855; Cell Signaling Technology, Danvers, MA, USA), phospo-Parkin (PA5-114616; Thermo Fisher Scientific), Atg7 (10088-2-AP; Proteintech Group, Inc, Rosemont, IL, USA), and P62 (R31056; NSJ Bioreagents, San Diego, CA, USA) antibodies. All antibodies except Thy-1 and Prdx5 (1:200 dilution) were used in a 1:1000 dilution ratio. After washing steps, the membranes were incubated with horseradish peroxidase-conjugated anti-rabbit or mouse secondary antibodies at a 1:10,000 dilution (GeneTex) for o/n at 4 °C. The target bands were visualized with enhanced chemiluminescence solutions (Bio-Rad Laboratories, Hercules, CA, USA) and were detected using ImageQuant™ LAS 4000 (GE Healthcare).

### 2.8. Proteomics of EVs from hHPPSCs

The entire process of proteomics analysis is mentioned in the previous paper [11]. In briefly, proteomic analyses were carried out using R28 cells treated with PBS (control), R28 cells treated with CoCl_2_, and R28 cells treated with CoCl_2_ and HPPSCs_EVs to elucidate the large-scale effects of HPPSCs_EVs on undamaged cells. Protein digestion was carried out using filter-aided sample preparation (FASP) protocol [26] with Ultracel^®^ YM-30 centrifugal filters (Merck Millipore, Darmstadt, Germany). Sample analysis was performed using an LC-MS/MS system consisting of a Dionex Ultimate 3000 HPLC coupled with a Q Exactive™ Hybrid Quadrupole-Orbitrap MS (Thermo Fisher Scientific). Database search and data processing were conducted as previously reported [27]. Differentially expressed proteins (DEPs) were filtered with a cutoff *p*-value ≤ 0.05 and log2FC ≥ 1 (fold-change). Heatmap was generated by the Perseus software. Protein–protein interactions were analyzed using the String database (https://string-db.org/ (accessed on 13 October 2022)). Volcano plots were prepared using R version 3.6.1.

### 2.9. Construction of Optic Nerve Injury Model

The process of constructing a disease animal model is described in detail in a recent paper [1]. In brief, using ultra-fine self-closing forceps, the optic nerve was compressed at 2 mm site behind the globe for 5 s. Optic nerve compression (ONC) was performed in the left eye (oculus sinister; OS). Then the canthal incision was sutured. After thorough suturing of the canthal site, subtenon injection of HPPSC_EVs into the nasal side of the eyeballs of the rats was performed once. The rats were classified into the following groups: Sham (balanced salt solution (BSS) injection after optic nerve compression); hPSCs_EVs group (300 ug/0.06 mL injection after optic nerve compression); hHPPSCs_EVs group (300 ug/0.06 mL injection after optic nerve compression). The 4-weeks group animal tissue was used for analysis.

### 2.10. Flat-Mounted Retinas and RGC Survival Analysis

The expression of Brn-3a and Tuj1 in the retina of the rat was analyzed by the flat-mount technique, and the detailed method was described in the previous paper [1].

### 2.11. Statistical Analyses

All the results are presented as mean ± standard error of the mean (SEM). Data analyses were conducted using GraphPad Prism 9 software (GraphPad Software, Inc., La Jolla, CA, USA). Statistically significant differences for data analysis were described in figure legends.

## 3. Results

### 3.1. Characterization and Recovery Afforded by HPPSC_EVs

The expression levels of exosomal markers CD9, CD81, and CD63 were compared in EVs from hPSCs and hHPPSCs (Figure 1A). In hHPPSC_EVs, CD9 was more highly expressed than in PSC_EVs. To explore whether hHPPSC_EVs improved cellular recovery under hypoxic conditions, we exposed R28 cells to CoCl_2_ for 9 h, and then treated such hypoxically damaged cells with EVs. The hHPPSC_EVs significantly increased cell proliferation of damaged R28 cells by 8.7% (Figure 1B). In addition, the ATP level reduction during hypoxia was significantly restored (by 1.07-fold) (Figure 1C). Proteins, the expression of which was affected by CoCl_2_, approached normal levels. Hif-1α expression increased after CoCl_2_ exposure but decreased significantly (by 0.79-fold) after hHPPSC_EV treatment (Figure 1D). By contrast, the expression of regeneration-related proteins, including vascular endothelial growth factor (VEGF), Thy-1, Gap43, Ermn, and neurofilament decreased under hypoxic conditions; the levels of VEGF, Thy-1, and neurofilament increased significantly by 1.80-, 1.98-, and 1.37-fold, respectively, after treatment with hHPPSC_EVs (Figure 1D). In particular, VEGF expression was increased by hHPPSC_EVs to a significantly greater extent than by PSC_EVs (1.26-fold) (Figure 1D).

### 3.2. Hierarchical Clustering and Gene Ontology

We subjected R28 cells to proteomic analysis. Hierarchical clustering of differentially expressed proteins (DEPs) was determined under three conditions (control, CoCl_2_, and CoCl_2_ + HPPSC_EVs) (Figure 2A). The Venn diagram in Figure 2B shows the expression levels of proteins associated with injury and recovery. The control, CoCl_2_, and CoCl_2_ + HPPSC_EVs groups expressed 1887, 1704, and 2057 proteins, respectively; the profile changed after exposure to CoCl_2_ (Figure 2B). Proteins unique to both R28 cells damaged by CoCl_2_ and cells exposed to CoCl_2_ + HPPSC_EVs were principally those of complexes and RNA-binding proteins. An interaction was evident among LONP1; PARK7; HSPD1; VDAC1, 2; and 3. This interaction and those of other proteins involved in mitochondrial function are shown in Figure 2C.

### 3.3. Volcano Plot of the Proteins of HPPSC_EVs Acting on Hypoxia-Damaged Retinal Precursor Cells

We drew volcano plots to identify changes in protein expression when hypoxic cells were exposed to HPPSC_EVs (Figure 3). Proteins that were up- and down-regulated are listed. Lonp expression was significantly decreased by CoCl_2_ (compared to the control) (Figure 3A). HPPSC_EV treatment restored LONP protein expression (Figure 3C). In addition, HSPA9, a mitochondrial 70 k Da heat shock protein (mtHsp70), was maintained in a high state in groups treated with HPPSC_EVs and CoCl_2_ (Figure 3A,B). Interestingly HPPSC_EVs treatment to damaged cells decreased HSPA9 expression (Figure 3C). The top 35 up- and down-regulated DEPs of each group are presented in Figure 3.

### 3.4. Role of LONP1 and the Pink/Parkin/p62 System during Recovery of Mitochondrial Function Induced by HPPSC_EVs

The results above suggest that Lonp1 played a role in mitochondrial quality-control by HPPSC_EVs in a hypoxic environment. Therefore, we evaluated how the lack of Lonp1 protein-affected R28 cells damaged by CoCl_2_. As shown in Figure 4, the expression of the mitophagy-associated proteins Pink1, Phospho-Parkin, Atg7, p62, Lc3b, Mfn2, and Mul1 was reduced significantly in CoCl_2_-damaged R28 cells. Although all tended to increase after exposure to HPPSC_EVs, only the p62 and Lc3b levels were significantly restored (by 1.21- and 1.40-fold, respectively) (Figure 4). When Lonp1 levels were low, the levels of the mitophagy-related proteins Pink1, Atg7, and Lc3b were significantly increased. However, CoCl_2_ treatment inhibited expression of these proteins in both normal and Lonp1 knockdown cells, and HPPSC_EVs did not rescue expression (Figure 4), suggesting that Lonp1 could play a crucial role in mitophagy.

### 3.5. HPPSC_EVs Increase Regeneration of Hypoxia-Damaged Retinal Precursor Cells

The level of Bnip3 was raised sharply in CoCl_2_-induced hypoxic condition. HPPSC_EVs lowered these levels with or without Lonp1 (Figure 5A). Interestingly, Vegf was only significantly recovered (1.47-fold) by HPPSC_EVs in scrambled cells but not in Lonp1-deficient cells. Among these neuronal markers, Gap43 was significantly decreased in lonp1-lacking protein (Figure 5A). From this data, we assumed that Vegf is a major target for regeneration.

### 3.6. Effects of HPPSC_EVs on Mitochondrial Quality Control

Next, we measured ATP levels (Figure 5B). CoCl_2_-induced hypoxia significantly reduced ATP production (Figure 5B) and increased the levels of reactive oxygen species (ROS) (Figure 5D). These abnormalities were rescued by HPPSC_EVs (Figure 5D). However, as above, HPPSC_EVs did not function in the absence of Lonp1. Next, we measured the levels of mitochondrial ATP synthase, Atp5a, and the antioxidant protective protein Prdx2. HPPSC_EVs significantly rescued Atp5a and Prdx2 expression (by 1.54- and 1.33-fold, respectively) after reduction by CoCl_2_, but only when Lonp1 was expressed (Figure 5C,E).

### 3.7. Effects of HPPSC_EVs in an In Vivo Model of Hypoxic Damage

#### 3.7.1. Changes in Neurogenic Marker Expression after Injection of HPPSC_EVs in an Optic Nerve Compression Animal Model

To assess RGC regeneration, we counted the numbers of rat RGCs stained with Brn-3a and Tuj1 (Figure 6A). After ONC, we found that although PSC_EVs and HPPSC_EVs increased the expression levels of Brn-3a 35- and 56-fold and those of Tuj1 1.0- and 1.3-fold, respectively, only HPPSC_EVs significantly increased retinal Brn-3a and Tuj1 expression (compared to the age-matched sham-operated group) at four weeks (Figure 6A). Changes in the expression levels of Hif-1α, Vegf, Gap43, Ermn, and Neurofilament in the rat optic nerve were analyzed four weeks after ONC. The Vegf and Gap43 levels were significantly increased (1.45- and 1.68-fold, respectively) by HPPCS_EVs, which affected Vegf expression levels in a manner unlike that of PSC_EVs (Figure 6B). Both PSC_EVs and HPPSC_EVs significantly upregulated neurofilament levels (20.35- and 22.97-fold, respectively) after ONC (Figure 6B).

#### 3.7.2. Comparison of Mitochondrial Protein Expression between the Retina and Optic Nerve Tissue after ONC

We compared changes in the levels of mitochondrial homeostasis proteins in the retina and optic nerve tissue. One week after ONC, the retinal levels of Atp5a, Mfn2, Mul1, and Prdx2 were significantly increased by HPPSC_EVs (Figure 6C and Appendix A) but any other expression levels in the two- and four-week groups did not differ, with the exception of increased Atp5a expression levels in the four-week group. Unlike in the retina, the optic nerve Vegf, Mfn2, and Atp5a protein levels were significantly upregulated by HPPSC_EVs at four weeks (Figure 6C). The expression of Prdx2 in optic nerve increased from one week after HPPSC_EVs treatment (Figure 6C and Appendix A).

## 4. Discussion

Several therapies for neuronal injuries seek to restore or improve mitochondrial structure and function. Neurons are energy-demanding; mitochondria sustain the neuronal network and power communication by employing intricate molecular mechanisms of mitochondrial spread [20]. Small-molecule antioxidants, and electron donors or acceptors, protect against glaucoma and acute and ischemic optic nerve injury [21]. The antioxidant and anti-inflammatory mitochondrial effects of EVs from MSCs may aid treatment of myocardial infarction, spinal cord injury, retinal disease, and diabetes [28,29,30,31,32]. Increasing evidence indicates that MSC EVs (released either spontaneously or after cell activation) contain immunoregulatory and pro-regenerative factors. EVs (exosomes and microvesicles) contain proteins, lipids, and nucleic acids (of which the miRNAs have been widely studied) [33]. EVs are taken up by cells and release their contents (either in the cytoplasm or via interactions with receptors on target cells) to stimulate downstream intracellular pathways [32]. EVs are not self-replicating, and given their small size, can be sterilized via filtration, aiding therapeutic applications [34,35]. MSC-derived EVs modify the metabolism of target cells. EV cargoes reflect the MSC culture or environmental conditions and changes in energy metabolism.

We previously showed that EVs from placental MSCs were neuroprotective and promoted neurorecovery in vitro [11]. UBA2 expression increased in a time-dependent manner in response to hypoxia, and was upregulated by exosomes [1]. Hypoxic preconditioning upregulated MSC Hif-1α/VEGF levels [1]; MSCs thus treated engaged in Hif-1α/VEGF signaling that reduced cellular apoptosis, autophagy, and inflammation [1]. VEGF induction by MSCs mediated the differentiation of endothelial progenitor cells via a paracrine effect; anti-VEGF antibody inhibited such differentiation [1]. We thus hypothesized that EVs from hypoxically preconditioned MSCs might serve as a therapy for optic nerve injury.

However, the exact roles played by MSC-EVs on mitochondrial metabolism, tissue repair after injury, and prevention of apoptosis during ischemic stress or metabolic reprogramming, remain unclear [31]. MSCs may improve mitochondrial function and attenuate oxidative injury by inhibiting ROS production and enhancing mitochondrial dynamics. ROS levels are reduced by the antioxidants superoxide dismutase, catalase, and glutathione S-transferases and by the adjustment of the redox balance [32]. In terms of mitochondrial dynamics, EVs balance the syntheses of mitochondrial fusion genes [mitofusin (mfn1, mfn2) and optic atrophy 1 (Opa1)] that protect cells against environmental damage, enhance SIRT3 activation, and upregulate the peroxisome proliferator-activated receptor gamma coactivator 1-alpha (PGC-1α). Then mitochondrial biogenesis increases and mitochondrial shape, density, and mass are optimized [32].

EVs secrete cytokines and growth factors that upregulate the expression of anti-apoptotic proteins (BCL-XL, BCL-2) and downregulate the expression of pro-apoptotic proteins (BAX, BAK, BAD, and CYTC), thus minimizing mitochondrial injury [32]. The miRNAs of EVs target mitochondrial metabolic pathways after translocation into mitochondria. The “mitomiR” mitochondrial miRNA subset features both nuclear and mitochondrial transcripts [36]. Some mitomiRs specifically regulate mitochondrial metabolism (mir-149 controls PARP2 and SIRT1 expression, mir-326 regulates PKM2 expression in cancer cells, mir-25 controls Ca uptake and ROS production, and mir-128a regulates BMI-1 expression to ensure redox homeostasis) [32]. MiRNAs in MSC-EVs improve mitochondrial function in models of renal and cardiac disease [32]. In addition, various mesenchymal stem cells or endothelial progenitor cells-derived EVs could deliver mitochondrial transfer into damaged cell leading to restore bioenergetics according to the size of the EVs [37,38,39].

In this study, we found that HPPSC_EVs (170–180 nm in diameter)-stimulated ATP production and attenuated hypoxic injury of R28 retinal progenitor cells by activating the LONP1/p62 signaling pathway. In vivo, HPPSC_EVs significantly increased the levels of the antioxidants Prdx2 and Prdx5. That is, HPPSC_EVs repaired mitochondrial damage to neuronal tissue, and the effects were sustainable; however, further studies are needed.

EV treatment affords significant advantages compared to cell therapies. EVs do not replicate, change their phenotype, or migrate from an application site; manipulation is accurate [40]. Dosages can be precise, as EVs do not divide. EVs are becoming recognized as useful biomarkers of neurodegenerative diseases. For example, spinal cord injury triggers differential regulation of the EV miRNAs that control calcium signaling, synaptic function, and axon guidance and degeneration [40,41]. Although the EV biology of the visual system is not well-characterized, recent studies have used EVs to detect and monitor optic nerve trauma and disease [42]. EVs from photoreceptors are highly expressed after rhegmatogenous retinal detachment [43], and EVs containing specific subsets of miRNAs serve as biomarkers for glaucoma detection and analysis [44].

RGCs are very vulnerable to mitochondrial dysfunction. Uniquely, their unmyelinated axons bend to exit the eye and become myelinated only in the optic nerve; the mitochondrial density of RGCs is the highest of all optic cells [21]. Sometimes, mitochondria are transported to regions of damage (such as axotomized axons); such mobilization correlates with axonal regeneration [22,23,24]. Axonal mitochondria are not cleared via localized mitophagy and exhibit dramatic retrograde transport during mild stress [25].

LONP of the DEPs and new markers of the HPPSC_EV proteome may be involved in PINK/PARKIN mitophagy and the non-PARKIN, mitochondrial ubiquitin–proteasome system. We found that both the levels of Hif-1α and Lonp1 were increased in retina or optic nerve tissues from the same aged groups. Hif-1α, a regulator of the cellular response to hypoxia, is involved in cell survival, differentiation, and maintenance of homeostasis during hypoxia-injured cells [45,46], which also activates a specific cell-recovery process. Hif-1α controls the expression of VEGF to promote angiogenesis [47,48] or activates damaged cells undergo apoptosis through BNIP3, NIX, and NOXA [49,50], and induces mitophagy activating NIPE/NIX-dependent pathway and mitochondrial biogenesis [51,52,53]. However, the Lon protein is induced by Hif-1α and is required for the degradation of COX4-1 protein to retain mitochondrial respiration [54]. These processes protect against inordinate ROS production and damaged mitochondria.

However, our sample size was rather small; more samples must be subjected to high-throughput proteomic analysis. Data-independent acquisition methods improve proteomic evaluations and validate candidate marker proteins identified via relative protein quantification [55]. A proteome map and quantitative analysis using volcano plots may reveal new biological functions [55]. In terms of ubiquitin-related factors, we found that CoCl_2_ reduced the in vivo levels of UBA2, UBE2I, UBE2E3, and ubiquitin; the falls were rescued by EVs.

Hypoxic MSC preconditioning increased paracrine activity and chemokine receptor levels, with maintenance of the differentiation potential and cell homogeneity [17,18,19]. Although further studies are needed, hypoxic preconditioning seems to enhance therapeutic MSC effects on neurological disorders and traumatic injuries. Our method aids the application of stem cell therapies in real clinical situations. When using subtenon hPSC injections to treat damaged optic nerves, hypoxic preconditioning is recommended.

Our follow-up time after ONC was only four weeks; this was short but may be adequate. Clinically, any therapeutic effect of MSCs must be sustained; the duration of any neuroprotective effect on damaged neuronal tissue requires further examination. In addition, it is necessary to explore why hypoxic preconditioning enhances the effects of EVs.

We demonstrated that HPPSC_EVs can alleviate the disturbed mitochondrial homeostasis of RGCs exposed to hypoxic conditions through the LONP1 protein. We also found that HPPSC_EVs facilitate recovery of damaged axons in ONC. Thus, we expect that HPPSC_EVs could be a viable solution for the treatment of optic nerve injury.

## Figures and Tables

**Figure 1 cells-11-03720-f001:**
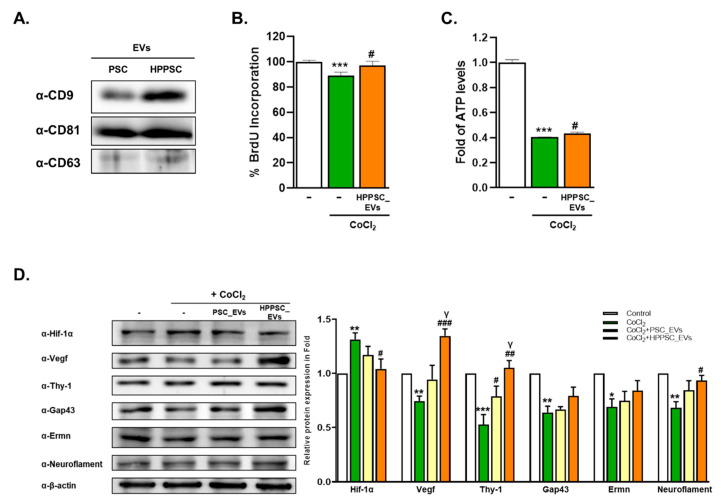
Characterization and recovery effects of hHPPSCs-derived EVs. (**A**) Isolated EVs expressing classic exosomal markers CD9, CD81, and CD63. R28 cells were treated with CoCl_2_ (200 μM). After incubation for 9 h, the cells were treated with EVs. (**B**) BrdU assays performed after 24 h. Data are presented as a mean ± SEM. (**C**) Determination of ATP production was presented as a fold (mean ± SEM). (**D**) Western blot analyses of target protein expression levels, using R28 lysates with CoCl2. The levels of protein expression are quantified (bottom-panel). Significantly difference was estimated using an unpaired t test (* *p* < 0.05, ** *p* < 0.005, *** *p* < 0.001 vs. the control; ^#^ *p*< 0.05, ^##^ *p*< 0.01, ^###^ *p* < 0.005 vs. CoCl_2_; γp < 0.05 vs. PSC_EVs). All experiments were performed in triplicate.

**Figure 2 cells-11-03720-f002:**
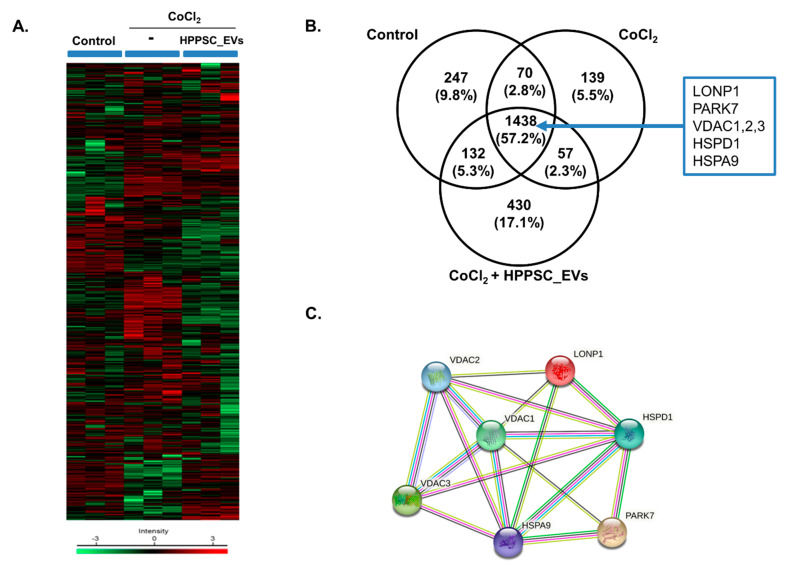
Cluster analysis of differentially expressed proteins (DFEs) by HPPSC_EVs in hypoxia-damaged R28 cells. MaxQuant version 1.5.8.3 (www.coxdocs.org (accessed on 13 October 2022)) for label-free quantification (LFQ) was used to identify differentially expressed proteins in treated groups (CoCl_2_, and CoCl_2_ + HPPSC_EVs) and PBS-treated control cells. (**A**) Heatmap of differentially expressed proteins (*p* ≤ 0.05 and log2FC ≥ 1) in lysates of three groups (Control, CoCl_2_, and CoCl_2_ + HPPSC_EVs) analyzed by hierarchical clustering. High expression is shown in red; low expression is shown in green. (**B**) Venn diagram showing the number of proteins identified in proteomic analysis of each group in R28 cells. (**C**) Identification of protein–protein interaction of DEF in experimental groups of R28 cells.

**Figure 3 cells-11-03720-f003:**
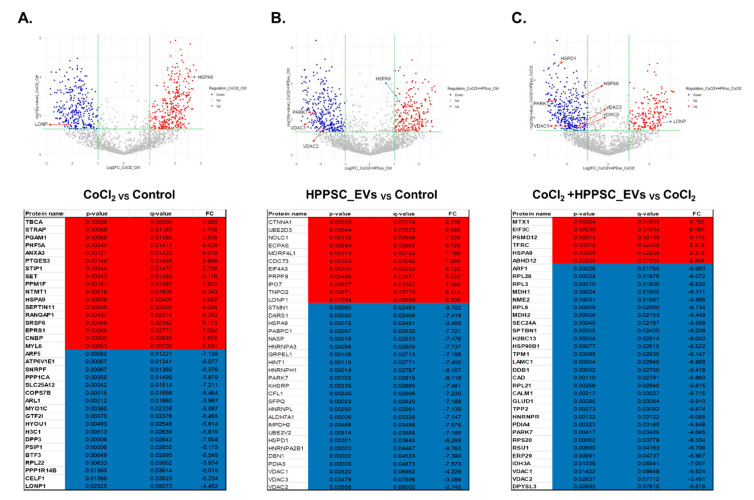
Identification of up/down-regulated DEPs in each group. Volcano plot of DEPs of each group. Red dots described up-regulated proteins and blue dots present down-regulated proteins. Proteins with significantly different expressions were presented.

**Figure 4 cells-11-03720-f004:**
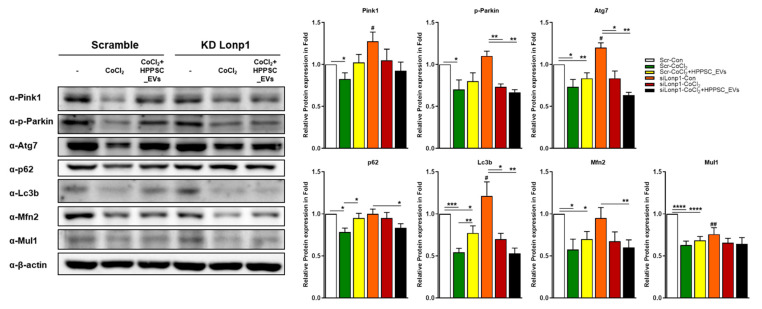
Involvement of Lonp1 during restoration induced by HPPSC_EVs. Scramble and siRNA targeting Lonp1 were transfected into R28 cells. After incubation, Lonp1 deficient cells were exposed to CoCl_2_ (200 μM). After incubation for 9 h, the hypoxia-induced cells were treated with HPPSC_EVs. Then, Western blot analyses were performed. The results are expressed as a mean ± SEM. Statistical significance was determined using a nonparametric statistical test, followed by the Mann–Whitney U test and an unpaired t test (^#^ *p* < 0.05,^##^ *p* < 0.01 vs. control with scramble, * *p* < 0.05; ** *p* < 0.005; *** *p* < 0.0005; **** *p* < 0.0001). All experiments were performed in triplicate. The KD experiments were performed five times.

**Figure 5 cells-11-03720-f005:**
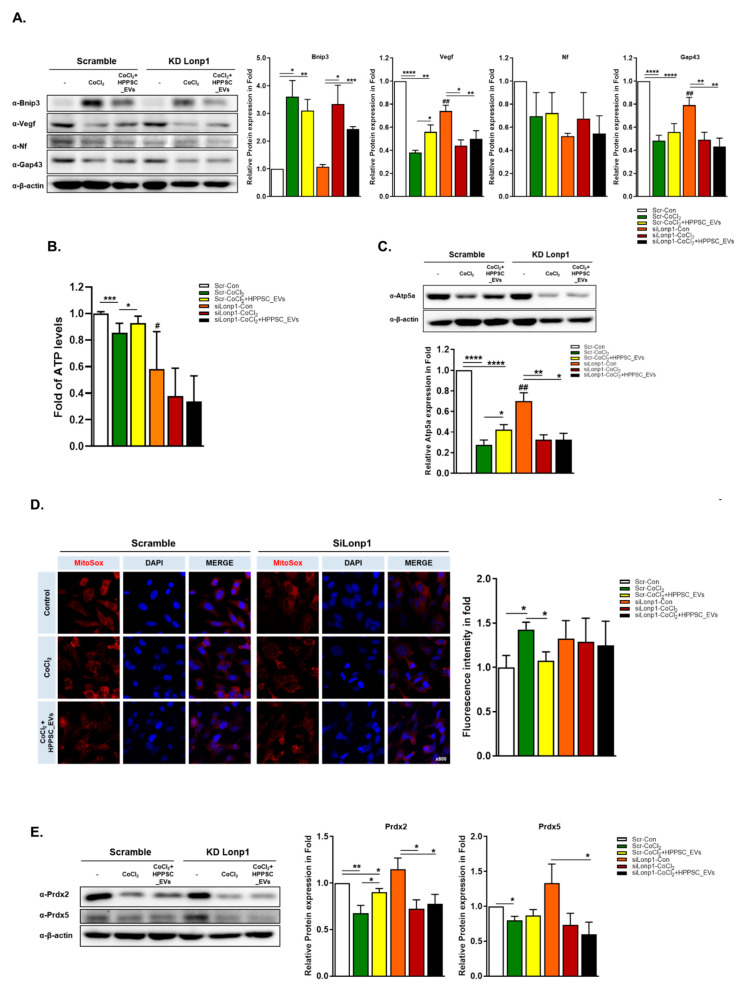
HPPSC_EVs regulate regeneration and mitochondrial function in the retinal precursor cells. Levels of regeneration-related proteins were determined by immunoblot analyses of hypoxia-damaged R28 cells. (**A**) Quantified values of Bnip3, Vegf, Neuroflament, NeuN, and Gap43 expressions are presented (right panel). (**B**) The changes of ATP level and (**C**) Atp5a protein, mitochondrial marker, expression by HPPSC_EVs in Lonp1-deficient cells and were presented as a fold (mean ± SEM). (**D**) Representatice imaged of damaged R28 cells stained with MitoSox. R28 cells were exposed to CoCl_2_ (200 μM) for 9 h and then treated with HPPSC_EVs (12 μg/mL). Using ZEN software, the intensity was measured. (**E**) The expression of antioxidant enzymes (Pdrx2 and Prdx5) also was determined in without Lonp1 protein cells. Lonp1 knock-down cells were treated with CoCl_2_ (200 μM). After incubation for 9 h, the hypoxia induced cells were incubated with HPPSC_EVs. After 24 h, the cells were harvested for immunoblot. The levels of protein expression are quantified (right-panel). Significantly difference was estimated using an unpaired *t* test (* *p* < 0.05, ** *p* < 0.005, *** *p* < 0.0005, **** *p* < 0.00005, ^#^ *p*< 0.05, ^##^ *p*< 0.01). KD experiments, ATP, and ROS measurement analysis were repeated three times.

**Figure 6 cells-11-03720-f006:**
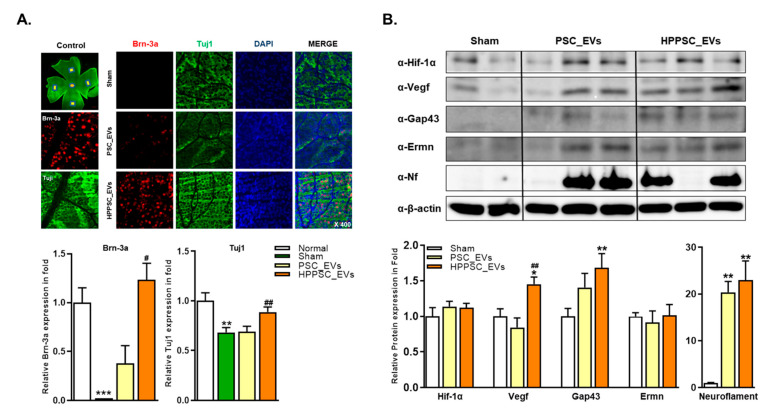
Effects of HPPSC_EVs on recovery and regeneration of RGCs in ONC models. (**A**) Representative confocal microscope-based fluorescence images of Brn-3a and Tuj1 staining (original magnification: 400×) were presented in retinal tissues of optic compressed animal models injected with PSC_EVs and HPPSC_EVs. By immunoblot analysis, (**B**) regeneration related protein expressions were investigated in optic nerve tissues. And comparison of (**C**) mitochondrial function related protein expressions between retina and optic nerve tissues were performed. Total of two or three retinas and optic nerves from each group were used. The results are presented as a mean ± SEM. Significantly difference was estimated using an unpaired t test (* *p* < 0.05, ** *p* < 0.005, *** *p* < 0.0005, ^#^ *p*< 0.05, ^##^ *p*< 0.01 vs. the age-matched sham).

## Data Availability

Not applicable.

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
