# Peer review of "The Role of Extracellular Vesicles in Optic Nerve Injury: Neuroprotection and Mitochondrial Homeostasis"

_cells, 2022, doi:10.3390/cells11233720_

Round 1

Reviewer 1 Report

 In this study the authors show extracellular vesicles from human placenta-derived mesenchymal stem cells (hPSCs) under hypoxia condition restored proliferation and ATP content of CoCL2 treated R28 cells (retinal precursor cells), they also show proteome analysis on the R28 cells and find LONP1 is a key mediator of mitophagy and it involves in the restoration of mitochondrial function after hypoxia-induced optic nerve injury. Finally, they evaluate the neurogenic marker change and mitochondrial protein expression in the optic nerve compression animal model after injection of hPSC-derived extracellular vesicles. 

Overall the authors try to address the question about the role of extracellular vesicles from hPSCs in optic nerve injury and mitochondrial function using both in vitro cell line model and in vivo animal model. However, the following issues should be addressed: 

  1. English should be improved, there are many grammar mistakes across the paper. Some of the mistakes even affect reading. 

  2. Although the authors have shown exosomal markers, direct visualization of mitochondria is still essential. Mitotracker staining could be applied to visualize intracellular mitochondria as well as Fluorescence-activated Cell Sorting (FACS) of cell culture supernatant to visualize the distribution of extracellular vesicles. 

  3. A direct evidence to prove whether R28 cells internalize hHPPSCs-derived mitochondria. An experiment such as fluorescence labeled mitochondria from hHPPSCs and co-culture with R28 cells. 

  4. There is logic issue for the LONP1 knockdown experiments, the results showed that  HPPSC_EVs could not restore mitochondrial function in R28 cells when LONP1 is knocked down, however, HPPSC_EVs will bring fresh and healthy mitochondria into R28 cells. If these fresh and healthy mitochondria could not rescue the mitophagy-related proteins, it means there are other vesicles which are less important for the cells. More evidence and explanation is needed here. 

  5. Result #5 did not correlate any figures. Supposedly figure 5.

Author Response

1. English should be improved, there are many grammar mistakes across the paper. Some of the mistakes even affect reading. 

: Based on your advice, the paper was corrected with the professional English proof editing service and the certification number was also written at the end of the text again.

2. Although the authors have shown exosomal markers, direct visualization of mitochondria is still essential. Mitotracker staining could be applied to visualize intracellular mitochondria as well as Fluorescence-activated Cell Sorting (FACS) of cell culture supernatant to visualize the distribution of extracellular vesicles. 

: We performed Mitotracker staining of live cells and observed that mitochondria were packed in controls. As expected, CoCl2 caused mitochondrial fragmentation and HPPSC_EVs treatment reversed these changes induced by CoCl2. In addition, HPPSC_EVs treatment to healthy cells seemed like more abundant mitochondria density compared to the controls.

3. A direct evidence to prove whether R28 cells internalize hHPPSCs-derived mitochondria. An experiment such as fluorescence labeled mitochondria from hHPPSCs and co-culture with R28 cells. 

: I deeply agree with your point of EV-derived mitochondria effect. In recently paper, it was demonstrated that medium to large EVs (m/lEV) contains mitochondria (Kandarp M. Dave et al, doi.org/10.1101/2021.10.29.466491).

Therefore, we stained Dil and Mitotracker with EVs used in our study. We couldn’t see the stained mitochondria in the HPPSC_EVs treated R28 cells. Because the size of EVs used in our study was about 170-180 nm (it was additionally described in M&M). Based on the reference, small EVs did not contain mitochondria. In addition, EVs are too small to visualize in current our imaging system using confocal microscopy. (Scale bar: 5μm)

4. There is logic issue for the LONP1 knockdown experiments, the results showed that HPPSC_EVs could not restore mitochondrial function in R28 cells when LONP1 is knocked down, however, HPPSC_EVs will bring fresh and healthy mitochondria into R28 cells. If these fresh and healthy mitochondria could not rescue the mitophagy-related proteins, it means there are other vesicles which are less important for the cells. More evidence and explanation are needed here. 

: We fully understand your reasonable concerns about the possibilities of the healthy mitochondria derived from HPPSC_EVs related with the mitophagy-related proteins. We additionally stained the Mitotracker and LC3B together and could find the improved LC3B expression in damaged R28 cells after HPPSC_EVs treatment.

As we presented in the answer to Question#3, they were mostly small-sized EVs used in our experiment. Even though it is hard to exclude the potential role of mitochondria derived from HPPSC_EVs, we have come to the conclusion that the mitophagy-related proteins were expressed from the recovered R28 cells by the HPPSC_EVs.

However, the further studies about the mitochondria tracking from the EVs are needed in the future. And these are mentioned in the Discussion. We appreciate your insightful comment to extend our scope and refine the paper.

5. Result #5 did not correlate any figures. Supposedly figure 5.

: I added figure number in result section. It was Fig. 5A.

Reviewer 2 Report

Dr. Park and the co-authors have evaluated the neuroprotective effect of extracellular vesicles (EVs) isolated from hypoxically preconditioned human placenta-derived mesenchymal stem cells (HPPSCs) using in vitro (rat R28 cell line) and in vivo (optic nerve crush) modes.

Although, this study reports several interesting observations, several issues should be addressed before this study is considered for publication:

1.     1.  It is not clear for how many passages human placenta-derived mesenchymal stem cells (hMSCs) were in culture before EV isolation.

2.     2.  The size distribution of EVs from control and treated hMSCs should be presented.

3.      3. A short description of the optic nerve injury model would be useful.

4.     4.  Author should also include the route and time of delivery of EVs in vivo after ONC. The number of exosomes versus protein concentration would be more informative.

5.     5.  In several cases it is not clear how many times the experiments have been repeated. It is not clear if they represent biological replicates. These data should be provided for all figures.

6.     6.  It is not clear why the authors stated that “An interaction was evident among LONP1, PARK7, HSPD1, and VDAC1, 2, and 3.” Was this statement based just on the String database?

7.      7. To assess RGC regeneration, the authors counted the number of rat RGCs stained with Brn3a. It is known that the expression of Brn3a is downregulated after retinal damage. It would be more appropriate to estimate the number of RGCs using RBPMS immunostaining.

8.     8.  It would be helpful to provide some explanations why two different conditions were used for creating hypoxia in HPPSCs) and R28 cells.  

Author Response

1. It is not clear for how many passages human placenta-derived mesenchymal stem cells (hMSCs) were in culture before EV isolation.

: I added passage numbers of HPPSCs in M&M.

2. The size distribution of EVs from control and treated hMSCs should be presented.

: The size of EVs used in our study was about 170-180 nm. And I added information of EVs in M&M.

3. A short description of the optic nerve injury model would be useful.

: I added in M&M.

4. Author should also include the route and time of delivery of EVs in vivo after ONC. The number of exosomes versus protein concentration would be more informative.

: The injection of EVs was performed one time after ONC. After 4 weeks, animal tissue was used for analysis. Using the Exo-quick kit, we obtained an average amount of 1.76 x 1010 / 3.48 mg EVs. For the injection of the same volume in all groups, we choose a protein enrichment method.

5. In several cases it is not clear how many times the experiments have been repeated. It is not clear if they represent biological replicates. These data should be provided for all figures.

: I revised in the Figure legends.

6. It is not clear why the authors stated that “An interaction was evident among LONP1, PARK7, HSPD1, and VDAC1, 2, and 3.” Was this statement based just on the String database?

: We found Lonp1, Park7, HSPD1, and VDAC in proteomics analysis. Although in this study, we presented only mitophagy-related signals by EVs, the next step is to study interaction networks associated with EVs. So I mentioned the candidate proteins found in proteomics in the abstract.

7. To assess RGC regeneration, the authors counted the number of rat RGCs stained with Brn3a. It is known that the expression of Brn3a is downregulated after retinal damage. It would be more appropriate to estimate the number of RGCs using RBPMS immunostaining.

: I deeply respect your critical point regarding the analyzing RGCs. We also considered RBPMS staining for RGCs, but existing Brn3a counting routines cannot be easily adapted to RBPMS labelling due to the nature of labelling: nuclear (Brn3a) versus cytoplasmic (RBPMS) staining.

Furthermore, counting of cytoplasmic RGC labelling is complicated since the fact that the many RGCs subtypes cover a wide range of soma sizes, from 10 to 35 µm in diameter.

Therefore, we thought that the brn-3a counting would be more accurate in terms of indicating RGCs after damage. We are willing to perform double staining Brn-3a and RBPMS in retina in next research, as you recommended.

8. It would be helpful to provide some explanations why two different conditions were used for creating hypoxia in HPPSCs) and R28 cells.  

: As for the clinical application of EVs or PSCs as the therapy in the optic nerve injury, we should consider the chemical toxicity of CoCl2. Thus, we had to choose two different methods for hypoxic condition. In addition, to construct efficient and stable damaged model, we used CoCl2 chemical for hypoxic condition.

Round 2

Reviewer 1 Report

The authors addressed my concerns and questions properly. It can be accepted in present form. 

Author Response

We appreciate your kind consideration on this paper.

Reviewer 2 Report

The authors addressed most of my comments, but I think that more information should be provided about the size distribution of EVs from control and treated hMSCs. It is not clear how it was evaluated and why the size of EVs (about 170-180 nm) was bigger than the size of EVs produced by such cells and described in the literature. It would be helpful to present a size distribution of isolated EVs.  

Author Response

: For isolation of EVs, I used ExoQuick-TC kit. ExoQuick is made with proprietary polymer that gently precipitates exosomes and microvesicles between 30 and 200 nm in size.

EVs isolated by ultracentrifugation were classified into medium to large EVs (m/lEV, 100-1000 nm) or small EVs (sEV, 30-200 nm) in the other study (Kandarp M. Dave et al, doi.org/10.1101/2021.10.29.466491).

I assumed that these polymer precipitation system or cell type could decide the size of EVs. The EVs used in experiment would be classified as small EVs. The size of EVs was evaluated by PMX-120 ZetaView® Mono Laser. This equipment was mentioned in M&M section.

We appreciate your kind consideration on this paper.
